

# RNA sequencing-based exploration of the effects of far-red light on microRNAs involved in the shade-avoidance response of *D. officinale*

Yifan Yang[1], Yuqiang Qiu[2], Wei Ye[3], Gang Sun[4] and Hansheng Li[1]

[1] College of Architectural Engineering, Sanming University, Sanming, China
[2] Xiamen Institute of Technology, Xiamen, China
[3] The Institute of Medicinal Plant, Sanming Academy of Agricultural Science, Sanming, China
[4] College of Resources and Chemical Engineering, Sanming University, Sanming, China

Corresponding author
Hansheng Li, lhs9897@163.com

## ABSTRACT

*Dendrobium officinale* (*D. officinale*) has remarkable medicinal functions and high economic value. The shade-avoidance response to far-red light importantly affects the *D. officinale* productivity. However, the regulatory mechanism of miRNAs involved in the far-red light-avoidance response is unknown. Previous studies have found that, in *D. officinale*, 730 nm (far-red) light can promote the accumulation of plant metabolites, increase leaf area, and accelerate stem elongation. Here, the effects of far-red light on *D. officinale* were analysed *via* RNA-seq. KEGG analysis of miRNA target genes revealed various far-red light response pathways, among which the following played central roles: the one-carbon pool by folate; ascorbate and aldarate; cutin, suberine and wax biosynthesis; and sulfur metabolism. Cytoscape analysis of DE miRNA targets showed that novel_miR_484 and novel_miR_36 were most likely involved in the effects of far-red light on the *D. officinale* shade avoidance. Content verification revealed that far-red light promotes the accumulation of one-carbon compounds and ascorbic acid. Combined with qPCR validation results, the results showed that miR395b, novel_miR_36, novel_miR_159, novel_miR_178, novel_miR_405, and novel_miR_435 may participate in the far-red light signalling network through target genes, regulating the *D. officinale* shade avoidance. These findings provide new ideas for the efficient production of *D. officinale*.

## INTRODUCTION

*Dendrobium officinale* Kimura et Migo (*D. officinale*) is an herbal medicinal plant species of the Orchidaceae family. Modern pharmacological studies have shown that the main functional metabolites of this species include polysaccharides, flavonoids, bibenzyl and alkaloids (*Zhan et al., 2020*). *D. officinale* has substantial medicinal functions and high economic value, and global demand is increasing annually. As such, researchers have

used different methods to increase the yield and content of the medicinal components of *D. officinale*, and light regulation is one of the important methods (*Xu, Gao & Ruan, 2015*).

Seven hundred-thirty–nanometre far-red light has a shade-avoidance effect on plants (*Chen et al., 2015*). The shade-avoidance response of most plants is manifested through a series of unique morphological changes and characteristics. For example, leaf area increases, plant height increases, photosynthetic physiological characteristics change, and dry matter accumulation in the stems increases (*Yang et al., 2017a*; *Yang et al., 2017b*; *Schambow, Adjesiwor & Lorent, 2019*). The plant shade-avoidance response can also regulate the growth of plants, resulting in earlier flowering, shorter vegetative growth time, and faster reproductive growth (*Schambow, Adjesiwor & Lorent, 2019*). In chrysanthemum (*Dendranthema morifolium* (Ramat.) Tzvel.), leaf chlorophyll fluorescence parameters (*i.e.,* Fv/Fm and qP) showed a trend of first increasing and then decreasing with increasing red:far-red (R:FR) (*Zhang et al., 2012*).

Important progress has been made in understanding the molecular mechanism of the far-red light response of plant shade avoidance. The plant shade-avoidance response is mainly mediated by the phytochrome PHYB, which acts by regulating the expression of downstream genes through the phytochrome-acting factor (PIFs) (*Zhan et al., 2020*; *Amanda, Herrera & Maloof, 2016*). For example, under high R:FR conditions, PHYB is activated to bind and phosphorylate PIF4 and PIF5 for degradation by the 26S proteasome, and under low R:FR conditions, the phytochrome PHYB is photoconverted to an inactive Pr configuration and exported from the nucleus, thereby enhancing the stability of PIF4 and PIF5 proteins, which in turn promotes the expression of genes that regulate stem elongation. The molecular mechanism of the plant shade-avoidance response may also be comediated by phytochromes and PIF/DELLA interactions. For example, under low R:FR conditions, PHYB is converted into an inactive Pr configuration, releasing PIF4 and PIF5 and allowing them to return to the nucleus. The stability of DELLA proteins decreases, and the binding ability to PIF4 and PIF5 is also reduced. However, the activity of PIF5 increases, which promotes the expression of downstream shade-avoidance response genes in plants and promotes plant elongation. Plants may also avoid shade through cryptochrome signalling. Relevant studies have shown that cryptochrome mainly mediates the shade-avoidance response through the CRY-SPA1/COP1 pathway and inhibits the shade-avoidance response of plants by preventing the degradation of the positive regulators of photomorphogenesis, such as HY5 and HFR1 (*Lyu et al., 2021*).

Some progress has been made in understanding the molecular mechanism of plant light-responsive miRNAs, but there have been no reports on the involvement of miRNAs in the shade-avoidance effect of far-red light on *D. officinale*. By regulating the expression of target genes, noncoding RNAs play an important role in plant growth and development, light signal transduction, stress resistance, epigenetic phenomena and other processes (*Byeon, Bilichak & Kovalchuk, 2018*; *Häfner, Talvard & Lund, 2017*; *Lyu et al., 2021*). According to the length of their nucleotides, noncoding RNAs can be divided into two categories: small RNAs (sRNAs) and long noncoding RNAs (lncRNAs). sRNAs are a class of noncoding RNAs with a length of approximately 22 nt; these RNAs mainly include microRNAs (miRNAs) and short interfering RNAs (siRNAs). In recent years, increasing amounts

of evidence suggest that miRNAs play an important role in plant photomorphogenesis. *Zhou, Fan & Li (2016)* found that when dark-grown Chinese cabbage seedlings were exposed to blue light or UV-A, the abundance of miR156 and miR157 decreased, thereby reducing the inhibition of the target genes *SPL* 9 and *SPL* 15, which in turn affected the photomorphogenesis of seedlings, resulting in short hypocotyls and expanded cotyledons. In the integrated analysis of potato miRNAs and transcriptomes by *Yan, Zhang & Zhang (2017)*, it was found that under light conditions, miRNAs could regulate the synthesis of potato alkaloids, lipid metabolism, and glycoalkaloid synthesis. In an miRNA omics study on the accumulation of functional metabolites in longan embryogenic calli in response to blue light, it was found that miR171 targets *DlDELLA*, that miR390 targets *DlBRI* 1, and that miR396 targets *DlEBF* 1/2 and *DlEIN* 3 and participates in the blue light signalling network, which in turn regulates the accumulation of functional metabolites in longan (*Li et al., 2018*).

The preliminary results of this project revealed that, in *D. officinale*, an appropriate proportion of 730 nm far-red light can promote an increased accumulation of secondary metabolites in plants, increase the area of leaves, and accelerate the elongation of stem segments, and plant productivity also improved (*Li et al., 2021*). In this study, we used high-throughput sequencing technology to identify putative miRNAs and investigated their expression profiles in *D. officinale* under far-red light conditions. By analysing the data of the control (CK) group and the light group, we identified the specific miRNAs involved in the far-red light on the shade avoidance of *D. officinale*, and the signal transduction pathway of these miRNAs involved in the shade-avoidance response of *D. officinale* was revealed. These results provide new ideas for the high-yield production of medicinal components in *D. officinale*.

## MATERIALS & METHODS

### Plant material and light treatment

The tissue culture-generated *D. officinale* seedlings selected in this project had 3-4 true leaves, their leaf width was approximately 2-3 mm, and their height was approximately two cm. The lighting conditions included red light (660 nm), blue light (450 nm), and far-red light (730 nm); the total light intensity was 200 $\mu$mol m$^{-2}$ s$^{-1}$, and total light duration was 60 d (12 h/d). The other conditions included a humidity of 55%–60% and a temperature of 25 $\pm$ 2 °C. The media in which *D. officinale* were cultivated consisted of 1/2-strength Murashige and Skoog (MS) media +30.0 g/L sucrose + 6 g/L agar + 1 g/L activated carbon (pH of 5.8). All the samples were flash frozen in liquid nitrogen and stored at −80 °C for nucleic acid extraction, sequencing and metabolite content determination. In this study, a group subjected to a red light intensity:blue light intensity:far-red light intensity ratio of 100:100:0 served as the CK group, and groups subjected to red light intensity:blue light intensity:far-red light intensity ratios of 80:80:40 (experimental group 2 (FR2)) and 40:40:120 (experimental group 8 (FR8)) served as the experimental groups. The FR2 *versus* CK, FR8 *versus* CK, and FR8 *versus* FR2 comparisons are denoted FR2-CK, FR8-CK, and FR8-FR2, respectively.

## Small RNA sequencing library construction

In this study, high-throughput sequencing was performed on the CK and light groups, with three biological replicates per treatment. The RNA samples were extracted with TRIzol (Thermo Fisher Scientific, Waltham, MA, USA). The purity, concentration and integrity of RNA samples were tested using advanced molecular biology equipment to ensure the use of qualified samples for transcriptome sequencing. Briefly, first, the 3′SR and 5′SR adaptors were ligated. Then, reverse transcription was performed to synthesize the first chain. Finally, PCR amplification and size selection were performed. A PAGE gel was used for electrophoresis fragment screening purposes, and rubber cutting was used to recycle the sRNA libraries. Finally, sRNA libraries were sequenced on an Illumina HiSeq 4000 platform (Hanzhou, China). All sequencing data of *D. officinale* under the different light treatments were deposited in the National Genomics Data Center (NGDC) Sequence Read Archive (accession number PRJCA010065).

## General analysis of sRNAs and prediction of miRNA targets

The clean reads were cleaned *via* sequence alignment of their sequences with those housed in the Silva database, GtRNAdb, Rfam database and Repbase database; ribosomal RNA (rRNA), transfer RNA (tRNA), small nuclear RNA (snRNA), small nucleolar RNA (snoRNA) and other ncRNA and repeats were removed. The remaining reads were used to detect known miRNA and novel miRNA predicted by comparing with genome and known miRNAs from miRBase. Randfold software was used for novel miRNA secondary structure prediction. TargetFinder software was then used to predict target genes according to the gene sequence information of known miRNAs, newly predicted miRNAs and miRNAs in corresponding species (*Allen et al., 2005*).

## Identification of differentially expressed (DE) miRNAs

Differential expression analysis of two conditions/groups was performed using the DESeq2 R package (1.10.1) (*Love, Huber & Anders, 2014*). DESeq2 provides statistical routines for determining differences in miRNA expression from digital data using a model based on the negative binomial distribution. The resulting $P$ values were adjusted using Benjamini and Hochberg's approach for controlling the false discovery rate. miRNAs with |log2[fold-change (FC)]| $\geq$ 0.58 and $P$ value $\leq$ 0.05 found by DESeq2 were considered DE. The power analysis is based on the method of *Hart et al. (2013)*. The power analysis in *D. officinale* under different light treatments is shown in Table S1.

## Identification of DE genes and functional annotations

The clean reads were mapped to the *D. officinale* reference genome using Bowtie v2.2.3 tools software (*Zhang et al., 2016*; *Langmead et al., 2019*). Prior to differential gene expression analysis, for each sequenced library, differential expression analysis of two samples was performed using edgeR (*Robinson, McCarthy & Smyth, 2009*). The $P$ value was adjusted using the $q$ value (*Storey & Tibshirani, 2003*). |log2($FC$)| $\geq$ 0.58 and $P$ value $\leq$ 0.05 were set as thresholds for significant differential expression. Gene functions were annotated based on the information within the following databases: the Nr (NCBI nonredundant protein

sequences); Kyoto Encyclopedia of Genes and Genomes (KEGG); Clusters of Orthologous Groups of proteins (KOG/COG); and Gene Ontology (GO) databases.

## Determination of functional metabolite contents

Folic acid determination: The folic acid content of *D. officinale* was measured using a commercial kit (Elk Biotechnology, Anhui, China) according to the manufacturer's instructions. Fine *D. officinale* stem and leaf powder (0.2 g) was dissolved in 50 µL of standard working solution and extracted for 1 h at 60 °C using an ultrasonic cleaning device. The samples were then centrifuged for 20 min at 1,000 × g. The supernatant was collected and assayed immediately. After the kit was equilibrated at room temperature, 50 µL of sample tissue was added to each well, 50 µL of biotinylated antigen working solution was immediately added to each well, after which the contents were mixed thoroughly and incubated at 37 °C for 60 min. The liquid in the plate was discarded, 200 µL wash buffer was added to each well, and the plate was washed three times. After drying, 100 µL of streptavidin-HRP working solution was added to each well and incubated at 37 °C for 60 min. The liquid in the plate was discarded, 200 µL of wash buffer was added to each well, and the plate was washed five times. After spin drying, 90 µL of TMB (3, 3′, 5, 5′-Tetramethylbenzidine) chromogenic substrate solution was added to each well and incubated at 37 °C for 20 min. The absorbance was measured with a UV-visible spectrophotometer (Evolution 350, Thermo Fisher, Waltham, MA, USA), and the wavelengths of folic acid was measured at 450 nm. The folic acid contents in the *D. officinale* were calculated according to established standard curves.

Ascorbic acid determination: The ascorbic acid content of *D. officinale* was measured using a commercial kit (Jiancheng Biotechnology Technology Co. Ltd., Nanjing, China) according to the manufacturer's instructions. First, 0.5 g of fine *D. officinale* powder was dissolved in 2.0 ml of plant protein extract. After water bath at 60 °C for 20 mins, the supernatant was collected by centrifugation at 12,000 rpm for 10 mins. 0.45 ml of reagent I was added to 0.15 ml of the supernatant, and mixed well by vortex. The solution was let stand for 15 mins and centrifuged at 4,000 rpm for 10 mins. Then, 0.5 ml of reagent II, 1.0 ml of reagent III, and 0.25 ml of reagent IV was added to 0.4 ml of the supernatant, mixed uniformly and heated in a water bath at 37 °C for 30 mins. Finally, 0.1 ml of reagent V was added to the solution, mixed well and let stand for 10 mins. The absorbance was measured with a UV-visible spectrophotometer (Evolution 350, Thermo Fisher, MA, USA), and the wavelengths of ascorbic acid was measured at 536 nm. The ascorbic acid contents in the *D. officinale* were calculated according to established standard curves.

## Quantitative real-time PCR (qRT–PCR) analysis

Total RNA from *D. officinale* was used for qRT–PCR validation of the mRNAs. Twelve miRNAs and their targets were subjected to qRT–PCR analysis on a LightCycler 480 Real-Time PCR System (Roche, Basel, Switzerland). cDNA synthesis, the reaction system, calculation methods and procedures, *etc.*, were the same as those of a previous method (*Li et al., 2022*). The Actin gene (NCBI accession number: JX294908) was used as reference gene (*Li et al., 2022*). The sequences of primers used are listed in Table S2.

## Data analysis

The test data of *D. officinale* were determined for at least 3 biological replicates. The data were analysed by Duncan's tests and one-way analysis of variance (ANOVA) through SPSS 19.0. Graphs were constructed *via* GraphPad Prism 6.0 software and OmicShare online software.

# RESULTS

## Global analysis of sRNA libraries from *D. officinale*

To explore miRNAs related to the shade-avoidance response of *D. officinale*, nine sRNA libraries (CK1, FR2, FR8) of plants under different light treatments were constructed and sequenced. After trimming adaptor sequences and filtering out corrupted adaptor sequences, remaining reads ranging from 18 to 30 nt were selected. These reads were clustered into unique sequences. In total, 14,759,803, 10,695,953, 15,029,297, 18,597,142, 18,935,396, 19,161,925, 18,434,038, 15,636,188 and 7,821,275 reads corresponding to 8,272,294, 6,256,825, 8,150,139, 10,332,462, 10,745,446, 10,492,521, 9,573,207, 8,371,783 and 4,518,335 genome mapped reads remained in the CK1, FR2 and FR8 libraries (Table 1).

Comparative analysis of CK1, FR2 and FR8 demonstrated that a number of sRNAs were DE in *D. officinale*. Then, the size distribution of sRNAs ranging from 18 to 30 nt was analysed across all libraries. In this study, the distribution of sRNAs under the different light treatments was similar, and the most abundant sRNAs were 24 nt in length, followed by those that were 23 nt, 22 nt, and 21 nt (Fig. 1).

## Differential expression analysis of *D. officinale* far-red light–responsive miRNAs

In this study, far-red light–responsive miRNAs were analysed in *D. officinale*, and the results showed that the miRNAs were regulated by different light combinations. A total of 65 DE miRNAs were identified in the FR2-CK combination, namely, 16 upregulated and 11 downregulated known miRNAs and 17 upregulated and 21 downregulated novel miRNAs (Fig. 2A and Table S3). In the FR8-CK comparison, 54 DE miRNAs were identified, namely, 19 upregulated and eight downregulated known miRNAs and 17 upregulated and 10 downregulated novel miRNAs (Fig. 2A and Table S3). Twenty-four miRNAs were coregulated in the FR2-CK and FR8-CK comparison groups (Fig. 2B and Table S4). A total of 40 DE miRNAs were expressed only in the FR2-CK combination; 19 were upregulated, and 21 were downregulated (Fig. 2B and Table S4). Moreover, a total of 29 DE miRNAs were expressed only in the FR8-CK combination; 22 were upregulated, and seven were downregulated (Fig. 2B and Table S4). The results of cluster analysis showed that the 50 known miRNAs DE in response to different light combinations could be divided into six groups on their basis of expression pattern differences (Fig. 2C and Table S5). The 69 novel miRNAs DE in response to the different light combinations could also be classified into six expression patterns (Fig. 2D and Table S6).

**Table 1  Summary statistics of sRNA libraries.**

| Library | Total reads | Mapped reads | Percentage (%) |
|---------|-------------|--------------|----------------|
| CK1 | 14,759,803 | 8,272,294 | 56.05% |
| CK2 | 10,695,953 | 6,256,825 | 58.50% |
| CK3 | 15,029,297 | 8,150,139 | 54.23% |
| FR2-1 | 18,597,142 | 10,332,462 | 55.56% |
| FR2-2 | 18,935,396 | 10,745,446 | 56.75% |
| FR2-3 | 19,161,925 | 10,492,521 | 54.76% |
| FR8-1 | 18,434,038 | 9,573,207 | 51.93% |
| FR8-2 | 15,636,188 | 8,371,783 | 53.54% |
| FR8-3 | 7,821,275 | 4,518,335 | 57.77% |

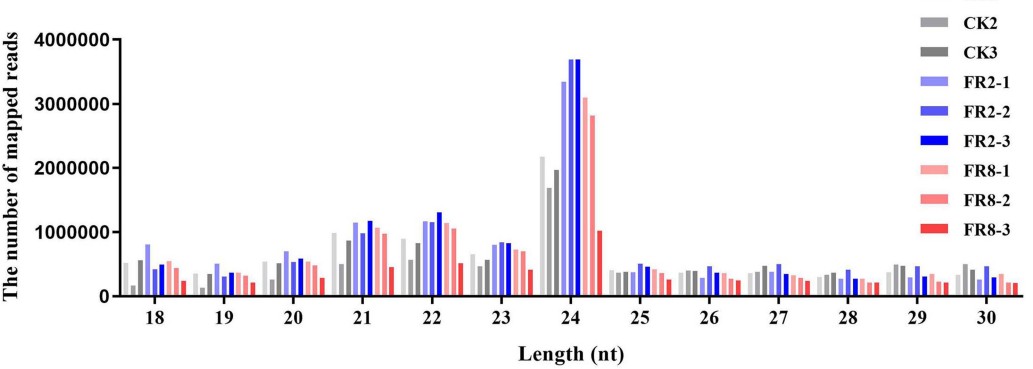

**Figure 1  Length distribution of sRNA sequences in the sRNA libraries.**

## Functional classification of conserved light-induced miRNAs in *D. officinale*

We used MapMan to perform KEGG enrichment analyses for the DE target genes (Fig. 3). The results showed that some pathways that were highly enriched and had a high number of genes. The FR2-CK combination mainly includes pathways involving SNARE interactions in vesicular transport, one-carbon pool by folate and sulfur metabolism. The FR8-CK combination mainly included one-carbon pool by folate, sulfur metabolism and mismatch repair. The FR8-FR2 combination mainly included one-carbon pool by folate, sulfur metabolism and mismatch repair.

In this study, the top 20 enriched pathways of three combinations (FR2-CK, FR8-CK, FR8-FR2) were analysed, and some pathways were enriched in the top 20 across the three combinations (Fig. 3D), such as one-carbon pool by folate, sulfur metabolism, mismatch repair, aminoacyl-tRNA biosynthesis, peroxisome and cutin, suberine and wax biosynthesis, showing that these pathways were significantly between the CK, FR2 and FR8 under the three light combinations.

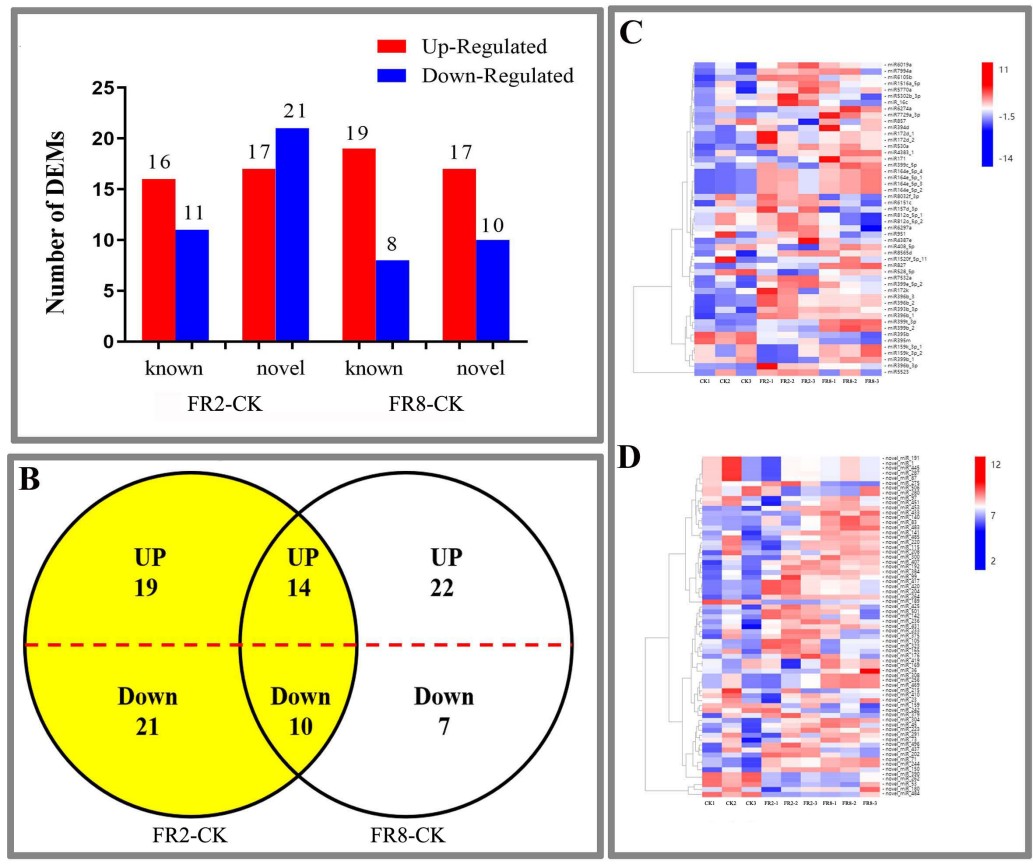

**Figure 2** **DE miRNAs in FR2-CK and FR8-CK.** (A) Number of miRNAs up- or downregulated in FR2-CK and FR8-CK; (B) Venn diagram showing the unique and common regulated miRNAs in FR2-CK and FR8-CK; (C) miRNAs DE in response to different light conditions. Included is a colour scale indicating the $\log_2$ (fold-change) in expression.

There were also some pathways enriched in the top 20 in two combinations. These pathways included glycerolipid metabolism, inositol phosphate metabolism, arachidonic acid, and diterpenoid biosynthesis, which were enriched in the top 20 in the FR2-CK and FR8-FR2 combinations (Fig. 3D). It is suggested that the light combination of FR2 may have a significant effect on these pathways. Protein export and glycosylphosphatidylinositol (GPI)-anchor biosynthesis were enriched in the top 20 pathways in FR2-CK and FR8-CK (Fig. 3D), indicating that SNARE interactions in vesicular transport, DNA replication and glycine, and serine and threonine metabolism under far-red light conditions may play an important role. Plant–pathogen interaction, circadian rhythm-plant, ribosome biogenesis in eukaryotes and ascorbate and aldarate metabolism were enriched in the top 20 in the FR8-CK and FR8-FR2 combinations (Fig. 3D), suggesting that the light combination of FR8 may have significant effects on these pathways.

Some pathways were enriched only in the top 20 in one combination; *e.g.*, phenylpropanoid biosynthesis and ubiquinone and other terpenoid-quinone biosynthesis were enriched only in the top 20 in FR2-CK (Fig. 3D), suggesting that the light combination

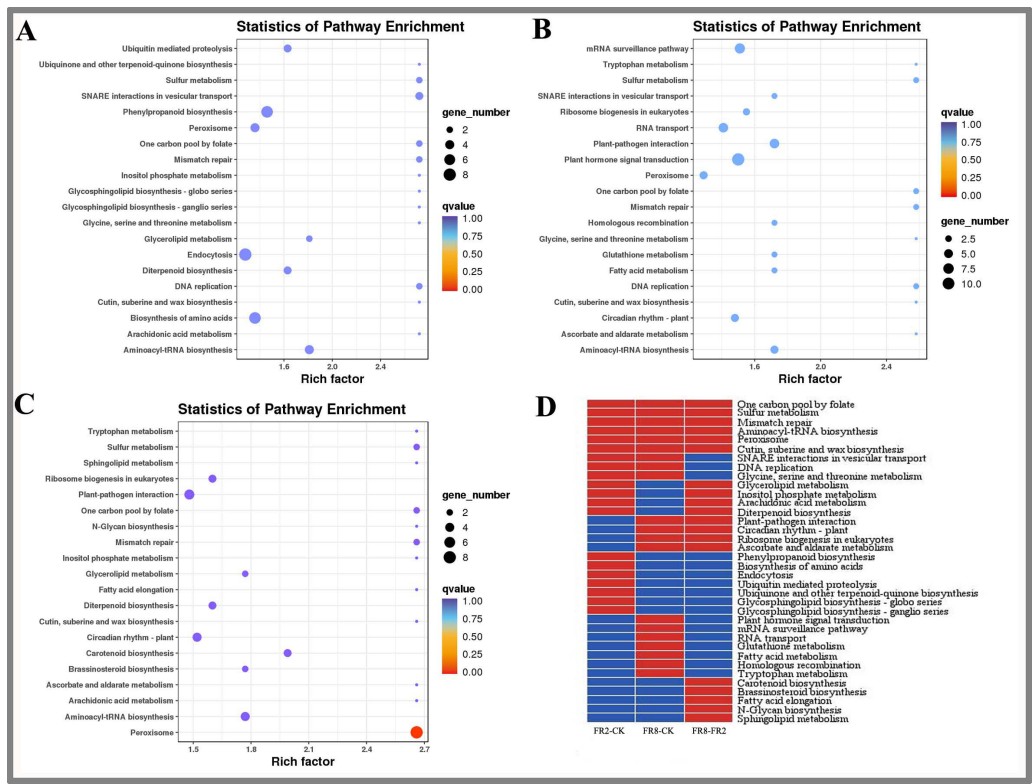

**Figure 3** **KEGG enrichment analysis of targets of DE miRNAs in _D. officinale_ under different lighting modes.** (A) FR2-CK. (B) FR8-CK. (C) FR8-FR2. (D) Top 20 KEGG pathways enriched in targets of DE miRNAs in the three groups. The red colour indicates that the comparison contains the pathway, and the blue colour indicates that the comparison does not contain the pathway.

of FR2 significantly affects those pathways. Plant hormone signal transduction, glutathione metabolism and tryptophan metabolism were enriched only in the top 20 in FR8-CK (Fig. 3D), indicating that the light combination of FR8 significantly affects these pathways. Carotenoid biosynthesis and brassinosteroid (BR) biosynthesis were enriched only in the top 20 in FR8-FR2, indicating that the FR8 light combination had a higher effect on carotenoid biosynthesis and BR biosynthesis than did the FR2 light combination (Fig. 3D).

## GO analysis of _D. officinale_ DE genes

To further understand the changes in the transcriptome of _D. officinale_ under far-red light culture, GO analysis was performed on the FR2-CK, FR8-CK, and FR8-FR2 combinations (Table 2).

In the FR2-CK comparison, GO analysis of biological processes showed that the biological process correlations of DE miRNA target genes included dephosphorylation, response to stimulus and response to stress. The FR8-CK combination mainly includes potassium ion transmembrane transport, intracellular signal transduction, cellulose biosynthetic processes and long-chain fatty acid metabolic processes. The FR8-FR2
**Table 2 GO term enrichment analysis of the target genes of DE miRNAs in *D. officinale* under different light treatments.**

| | ID | Description | Q value | | |
|---|---|---|---|---|---|
| | | | FR2-CK | FR8-CK | FR8-FR2 |
| Biological process | GO:0016311 | Dephosphorylation | 0.5739 | | |
| | GO:0050896 | Response to stimulus | 0.5739 | | |
| | GO:0006950 | Cellulose biosynthetic process | 0.5739 | 0.5617 | |
| | GO:0006950 | Response to stress | 0.5739 | | |
| | GO:0010501 | RNA secondary structure unwinding | 0.5739 | | |
| | GO:0071805 | Potassium ion transmembrane transport | | 0.5617 | |
| | GO:0016070 | RNA metabolic process | | 0.5617 | |
| | GO:0035556 | Intracellular signal transduction | | 0.5617 | 0.6027 |
| | GO:0001676 | Long-chain fatty acid metabolic process | | 0.5617 | 0.6027 |
| | GO:0043547 | Positive regulation of GTPase activity | | | 0.6027 |
| | GO:0009813 | Flavonoid biosynthetic process | | | 0.6027 |
| | GO:0006006 | Glucose metabolic process | | | 0.6027 |
| Cellular component | GO:0005829 | Cytosol | 0.6995 | | |
| | GO:0005783 | Endoplasmic reticulum | 0.6995 | 0.2106 | |
| | GO:0005747 | Mitochondrial respiratory chain complex I | 0.6995 | 0.7630 | |
| | GO:0005777 | Peroxisome | 0.6995 | | 0.7065 |
| | GO:0016021 | Integral component of membrane | 0.6995 | | 0.7065 |
| | GO:0009536 | Plastid | | 0.0523 | 0.7065 |
| | GO:0012505 | Endomembrane system | | 0.7630 | |
| | GO:0009506 | Plasmodesma | | 0.7630 | |
| | GO:0016020 | Membrane | | | 0.7065 |
| | GO:0005643 | Nuclear pore | | | 0.7065 |
| Molecular function | GO:0016887 | ATPase activity | 0.6407 | 0.6147 | |
| | GO:0016760 | Cellulose synthase (UDP-forming) activity | 0.6407 | 0.6147 | 0.6508 |
| | GO:0050662 | Coenzyme binding | 0.6407 | | |
| | GO:0005488 | Binding | 0.6407 | | |
| | GO:0004497 | Monooxygenase activity | 0.6407 | | |
| | GO:0003993 | Acid phosphatase activity | | 0.6147 | |
| | GO:0008017 | Microtubule binding | | 0.6147 | |
| | GO:0003700 | Transcription factor activity, sequence-specific DNA binding | | 0.6147 | |
| | GO:0019706 | Protein-cysteine S-palmitoyltransferase activity | | | 0.6508 |
| | GO:0004252 | Serine-type endopeptidase activity | | | 0.6508 |
| | GO:0016740 | Transferase activity | | | 0.6508 |
| | GO:0004672 | Protein kinase activity | | | 0.6508 |

combination mainly included intracellular signal transduction, long-chain fatty acid metabolic processes and flavonoid biosynthetic processes.

The most relevant cellular components of the FR2-CK combination included endoplasmic reticulum, mitochondrial respiratory chain complex I, peroxisome and integral component of membranes. The most relevant cellular components of the FR8-CK combination included the endoplasmic reticulum, mitochondrial respiratory chain

complex I, plastid, endomembrane system and plasmodesma. The FR8-FR2 combination included peroxisomes, integral components of the membrane and plastids.

In the FR2-CK combination, the GO analysis showed that the most relevant molecular functions of DE miRNA target genes were ATPase activity, cellulose synthase (UDP-forming) activity and monooxygenase activity. The FR8-CK combination mainly included ATPase activity, cellulose synthase (UDP-forming) activity and acid phosphatase activity. The FR8-FR2 combination mainly included serine-type endopeptidase activity and protein kinase activity.

In conclusion, the far-red light response to *D. officinale* shade avoidance may involve plant responses to external factors, potassium ion transmembrane transport, intracellular signal transduction, and metabolic processes.

### Network of miRNAs and far-red light–responsive targets

To further understand the function of DE miRNAs in the response of *D. officinale* to far-red light, a miRNA-target interaction network was constructed using Cytoscape software. In the FR2-CK combination, the number of target genes of novel_miR_484 reached 120; these were most likely involved in the effect of far-red light on the shade avoidance of *D. officinale*. Two, four, two, and two members of miR172s, miR395s, miR396s, and miR399s, respectively, have are in far-red light regulation. In addition, some target genes were found to be regulated by different miRNAs in FR2-CK. For example, the MA16_Dca025956 gene was coregulated by novel_miR_71 and novel_miR_244, and the MA16_Dca011199 gene was coregulated by novel_miR_99 and novel_miR_390 (Fig. 4A, Tables S7 and S9).

In the FR8-CK combination, novel_miR_36 had the most target genes and was most likely involved in the effects of far-red light on the shade avoidance of *D. officinale*. There are two and four members of miR395s and miR399s, respectively, involved in far-red light. In addition, some target genes were also found to be regulated by different miRNAs in FR8-CK. For example, the MA16_Dca009889 gene was coregulated by miR157d_3p and novel_miR_45, and the MA16_Dca018325 gene was coregulated by miR399c_5p and novel_miR_189 (Fig. 4B, Tables S8 and S9).

### Metabolite contents of *D. officinale* under different light conditions

In this study, the metabolites of *D. officinale* leaves and stem segments subjected to different far-red light were measured, and the results are shown in Fig. 5. The folic acid content of *D. officinale* leaves under the FR8 treatment was the highest (85.61 pg ml$^{-1}$), followed by that under the FR2 treatment (36.66 pg ml$^{-1}$), and that of the CK was the lowest—30.35 pg ml$^{-1}$ (Fig. 5A and Table S10). The folic acid content of stems under the FR8 treatment was the highest (88.67 pg ml$^{-1}$), followed by that under the FR2 treatment (69.95 pg ml$^{-1}$), and the lowest was in the CK—28.62 pg ml$^{-1}$ (Fig. 5B and Table S11). The leaves and stem segments under the FR8 treatment had the highest ascorbic acid content, followed by those under the FR2 treatment, and the lowest content was recorded in the CK (Figs. 5C–5D and Tables S12–S13).
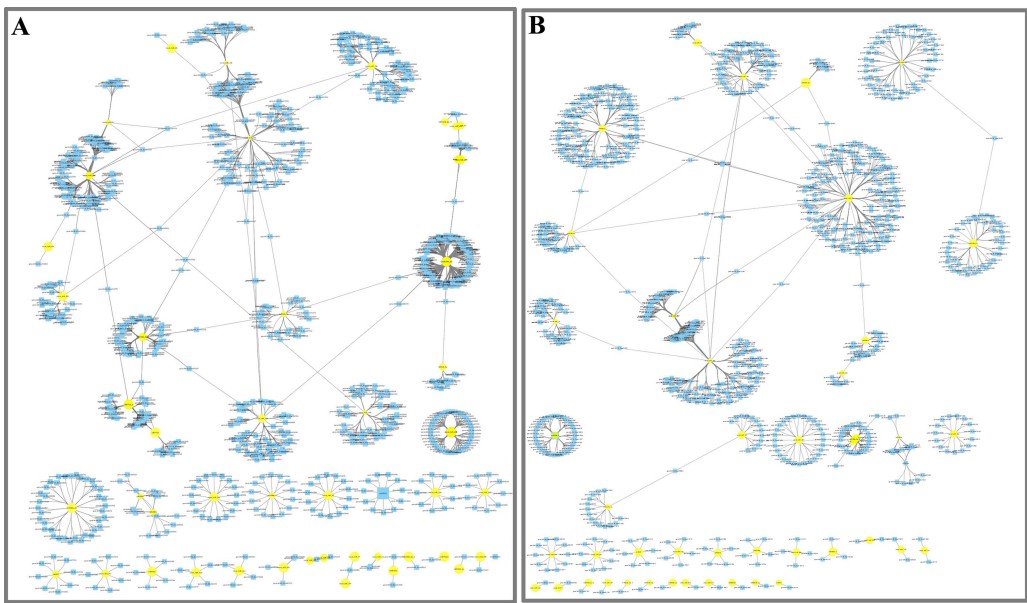

**Figure 4 Network analysis between miRNAs and far-red light–responsive targets.** (A) Interaction network of FR2-CK; (B) Interaction network of FR8-CK. The yellow diamond nodes represent miRNAs, and the blue square nodes represent mRNAs. High expression and low expression of miRNAs are indicated by the size of the diamonds. The solid lines indicate interaction associations between miRNAs and targets.

## MiRNAs and their targets in the far-red light signalling network in *D. officinale*

On the basis of previous studies on the signal transduction pathway of far-red light, by mining miRNA data, we found that novel_miR_36, miR395b, novel_miR_159, novel_miR_178, novel_miR_405, and novel_miR_435 could participate in the regulatory effects of far-red light on the shade-avoidance response of *D. officinale*. network. The target gene of novel_miR_36 is gene MA16_Dca006821, and its functional annotation was found to be phytochrome A-like (*PHY* A), which is significantly upregulated under far-red light. The target gene of miR395b was MA16_Dca015429, and its function was annotated as the transcription factor PIF3-like (*PIF* 3), which was significantly downregulated under far-red light. The target gene of novel_miR_159 is gene MA16_Dca005372, and its functional annotation is transcription factor PIF4. The target gene of novel_miR_178, novel_miR_405 and novel_miR_435 is gene-MA16_Dca020963, and its functional annotation is protein SUPPRESSOR OF PHYA-105 1-like (*SPA* 1).

## QPCR analysis of DE miRNAs and their target gene expression

Eleven miRNAs and target genes were verified by qPCR, and the results are shown in Fig. 6 and Tables S14–S15. The results showed that Novel_miR_53 targets gene-MA16_Dca007605, miR395b targets gene-MA16_Dca003285, novel_miR_36 targets gene-MA16_Dca020471, and novel_miR_159 targets gene-MA16_Dca005372. Only these four pairs of miRNAs and target gene expression patterns were negatively correlated (Fig. 6). This may be because a target gene is simultaneously regulated by different
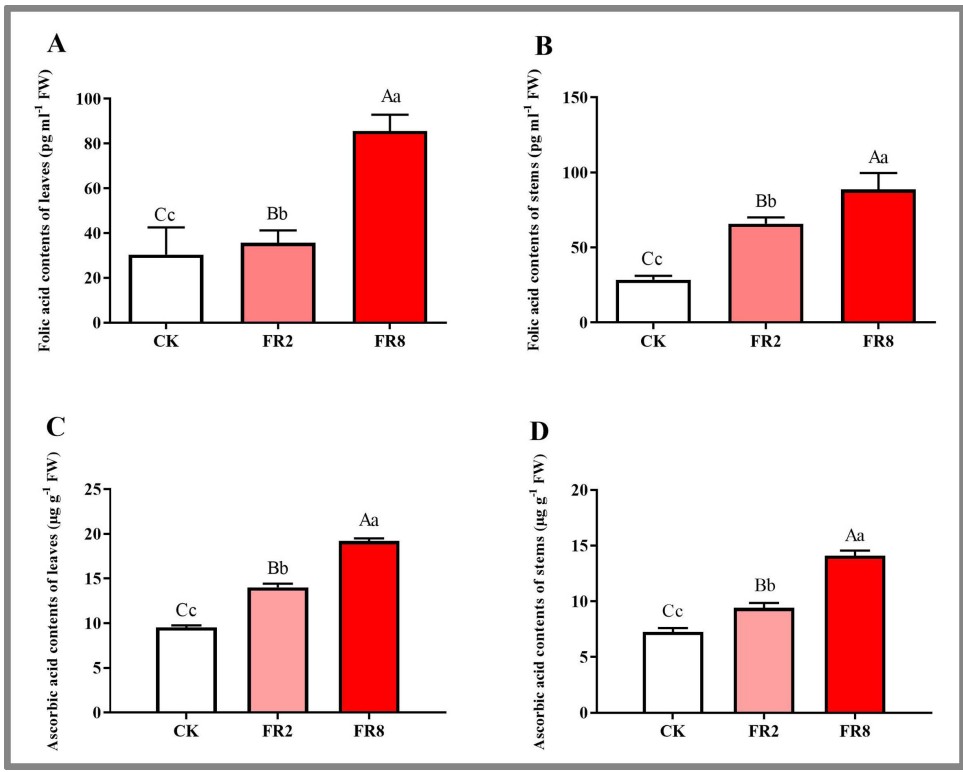

**Figure 5** **Metabolite contents in *D. officinale* under different light treatments.** (A and B) represent changes in the folic acid content in the leaves and stems, respectively. (C and D) represent changes in the ascorbic acid contents in the leaves and stems, respectively. The different upper/lowercase letters indicate statistically significant differences at the 0.01/0.05 level, respectively, as determined by one-way ANOVA and Duncan's test.

members of the miRNA family or other miRNAs in the plant response to changes in the light environment. When some miRNA members cannot regulate the expression of target genes, other members complement their functions to achieve the regulation of target gene expression levels. Some miRNAs can be involved in the regulation of sulfur metabolism, such as miR395b and the target gene MA16_Dca003285 (Fig. 6B). Some miRNAs can synthesize genes by targeting metabolic pathways, thereby regulating the accumulation of metabolites. For example, miR399c_5p targets the gene MA16_Dca022511 to regulate the ascorbic acid and uronic acid metabolic pathways, thereby affecting the accumulation of ascorbic acid and uric acid. The expression level of gene MA16_Dca022511 was the highest in FR8, followed by FR2, and was the lowest in CK, indicating that FR8 can promote the synthesis of ascorbic acid and uronic acid in *D. officinale* (Fig. 6C). Some miRNAs, such as miR399t_3p and its target gene MA16_Dca002672, were found to be involved in the biosynthesis of cutin, suberine, and wax (Fig. 6D). novel_miR_483 targets gene-MA16_Dca001413 to regulate potassium ion transmembrane transport (Fig. 6F), and novel_miR_390 targets gene-MA16_Dca000483 to regulate response to stimulus (Fig. 6G). There are also some miRNAs that, by targeting far-red light signalling network

Yang et al. (2023), *PeerJ*, DOI 10.7717/peerj.15001

13/26

genes (PHYA, PIF3, PIF4, SPA1), can affect the shade-avoidance response of *D. officinale* (Figs. 6H–6L).

## DISCUSSION

### miRNAs participate in the folic acid metabolic pathway and play a role in the shade-avoidance effect of far-red light on *D. officinale*

Among the top 20 KEGG pahways, one-carbon pool by folate was enriched in the FR2-CK, FR8-CK and FR8-FR2 combinations (Fig. 4). Physiological and biochemical experiments also verified that the far-red light treatment resulted in a higher wax content than did the white-light treatment (Fig. 1). Moreover, among the top 20, KEGG pathways, plant hormone signal transduction was enriched in the FR8-CK combination (Fig. 4).

The environment can affect the folic acid content of plants. Folic acid metabolism plays a key role in the plant stress response. Under salt stress, osmotic stress, drought stress, and oxidative stress of *Arabidopsis thaliana*, genes related to folic acid in, such as *AtDFD*, participate in C1 metabolism and participate in folic acid degradation. The expression levels of *AtGGH* 1, *AtGGH* 2, and *AtGGH* 3 in the apoplastic pathway increase, indicating that the environment can impact folic acid content (*Hanson, Beaudoin & Mccatry, 2016*). Different light quality, light intensity and air temperature will affect the folic acid content of vegetables during growth or storage after harvest (*Okazaki & Yamashita, 2019*). When irradiated with red light at 25 °C with a light intensity of 200 μmol m$^2$/s, lettuce presented the highest folate content. For lettuce, the highest folate content was found in the tested plants treated with 70% red light and 30% blue light in autumn and a combination of red and blue light in winter (*Dlugosz-grochowska, Oton & Wojciechowska, 2016*). In addition, folic acid metabolism is closely related to auxin (IAA) signal transduction, and IAA is involved in the plant response to shade avoidance. Relevant studies have shown that folic acid in the plant cytoplasm can inhibit the synthesis of starch in nonphotosynthetic cells, and the interaction between folic acid and sucrose can affect the sensitivity of plant seedlings to IAA and the distribution of IAA in plants (*Hayashi, Tanaka & Yamamoto, 2017*; *Stokes, Chattopadhyay & Wilkins, 2013*). Therefore, through various target genes, miRNAs may participate in the folic acid metabolism pathway, thereby affecting the shade-avoidance effect of far-red light on *D. officinale*.

### The *D.officinale* responds to far-red light *via* increased cutin, suberin and wax biosynthesis

Among the top 20 in KEGG pathways, cutin, suberine and wax biosynthesis were all enriched in FR2-CK, FR8-CK and FR8-FR2 combinations (Fig. 3). GO analysis of DE genes found that response to stimulus and response to stress were significantly enriched in the FR2-CK combination (Table 2). QPCR experiments also verified that the expression level of gene-MA16_Dca002672 (CYP86B1) in FR8 was higher than that of other treatments (Fig. 6D).

The cuticle of the plant covers the outermost layer of the plant, which is the direct contact surface between the plant and the environment and is also the first defence barrier of the plant. The stratum corneum is composed of cutin and wax, and the wax

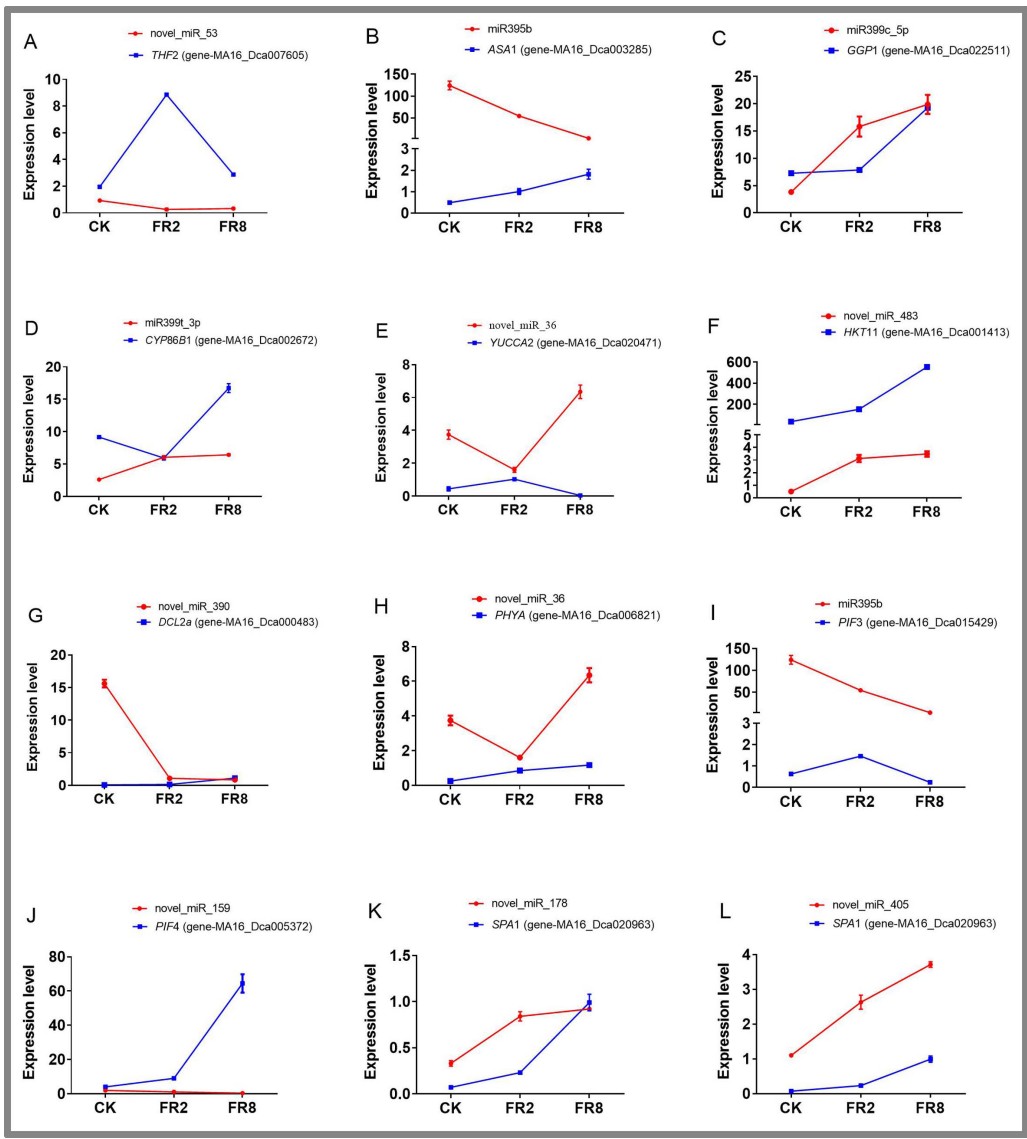

**Figure 6** **Identification of DE miRNAs and their targets in *D. officinale* under different far-red light conditions according to quantitative qPCR.** *THF* 2, formyltetrahydrofolate deformylase 2; *ASA* 1, ATP sulfurylase 1; *GGP* 1, GDP-L-galactose phosphorylase 1; *CYP* 86*B* 1, cytochrome P450 86B1; *YUCCA* 2, indole-3-pyruvate monooxygenase; *PPT* 11, probable potassium transporter 11; *DCL* 2a, endoribonuclease Dicer homologue 2a; *PHY* A, phytochrome A; *PIF* 3, phytochrome-interacting factor 3; *PIF* 4, phytochrome-interacting factor 4; *SPA* 1, PHYTOCHROME A SUPPRESSOR 1.

is composed of intrinsic wax and epidermal wax embedded in the keratin skeleton. Waxes play an important role in plant resistance to biotic and abiotic stresses; waxes inhibit nonstomatal water loss, reduce the retention of water on plant surfaces, protect plants from UV radiation, *etc.* (*Wang, Liu & Gai, 2015*). Studies have shown that stratum corneum wax deposition is sensitive to changes in the environment, such as drought, light, temperature, and humidity (*Li et al., 2022*; *Piroozian et al., 2016*; *Qiu et al., 2015*). Yellow light significantly increased the lamellar waxy crystal structure of the epidermis of faba bean leaves, and yellow light played a significant role in promoting the deposition of total wax and the deposition of dominant components such as primary alcohols and alkanes (*Lei et al., 2020*). Changes in the environment can not only affect the composition, content and crystal structure of stratum corneum wax but also affect the expression of wax-related genes. Transcriptome sequencing of *A. thaliana* revealed that genes related to wax biosynthesis and deposition, including *CER* 1, *LTP* 7, *LACS* 3, *LTP* 6, *LTP* 2 and *ABCG* 19, are regulated by the photoperiod (*Go, Kim & Kim, 2014*). Therefore, the cultivation of *D. officinale* with an appropriate amount of far-red light may promote the Cutin, suberine and wax biosynthesis of the plant and ensure normal growth and development.

## Important role of sulfur metabolism in the effects of far-red light on the shade-avoidance response of *D. officinale*

Among the top 20 KEGG pathways, sulfur metabolism and peroxisome were enriched in the FR2-CK, FR8-CK and FR8-FR2 combinations (Fig. 3). Moreover, among the top 20 KEGG terms, both ascorbate and aldarate metabolism were enriched in the FR8-CK and FR8-FR2 combinations (Fig. 3), and in the top 20 pathways, glutathione metabolism, plant hormone signal transduction, and tryptophan metabolism were enriched in the FR8-CK comparison (Fig. 3). Physiological and biochemical tests also verified that the content of ascorbic acid in the far-red light treatment was higher than that in the white-light treatment (Fig. 1). Both the target gene MA16_Dca003285 (*ASA* 1, ATP sulfurylase 1) of miR395b and the target gene MA16_Dca022511 (*GGP* 1, GDP-L-galactose phosphorylase 1) of miR399c_5p were expressed in the far-red light treatment compared with the white-light treatment (Fig. 6).

Sulfur metabolism plays a crucial role in the response of plants to light (*Li et al., 2019*). When plants respond to the environment, the sulfur transporters SULTR1;1 and SULTR2;1 are upregulated, and the activities of sulfur primary metabolism ATPS and OASTL are increased, which promotes the absorption and assimilation of sulfur (*Cao et al., 2014*; *Rodríguez-Hernández Mdel et al., 2014*). After the activity of assimilation pathway, inorganic sulfate is converted into cysteine, which together with glutamic acid and glycine forms glutathione. Glutathione is a part of the ascorbic acid-glutathione cycle (*Ahmad et al., 2016*). Glutathione reduces oxidized ascorbic acid, thereby scavenging the reactive oxygen species generated in plants due to the environment and thereby improving the adaptability of plants (*Ahmad et al., 2016*). When plants respond to the environment, the content of reactive oxygen species in throughout the plant body increases, and the reduced glutathione (GSH) and total glutathione also increase (*Ahmad et al., 2016*). There are also studies showing that the content of cysteine in the synthesis of glutathione is

much lower than that of glutathione, the synthesis of cysteine becomes a limiting element of glutathione, and cysteine is mainly composed of ATP sulfate (*Ahmad et al., 2016*). Cysteine biosynthesis is catalysed by enzymes such as O-acetyl serine lyase and O-acetyl serine lyase, which shows that the relationship between the environment and the primary metabolism of sulfur is very close (*Baig et al., 2019*). In addition, related studies have also found that sulfur metabolism is closely related to hormone signal transduction, and endogenous hormones (IAA, gibberellin (GA), ethylene (ETH), *etc.*) are all involved in the plant response to shade avoidance (*Koprivova & Kopriva, 2016*). *Wang, Waters & Smith (2018)* found that plant hormones are involved in the regulatory effects of $H_2S$ on seed germination. Low concentrations of NaHS are antagonistic to abscisic acid (ABA) and IAA, while high concentrations of NaHS are associated with GA, ETH, brassinosteroids (BRs), cytokinin (CTK) and salicylic acid (SA) antagonistically regulating seed germination. Further research showed that exogenous H2S may affect the germination of seeds by regulating the expression of related genes, altering the synthesis, metabolism and signal transduction of endogenous hormones in seeds. Therefore, during the process of *D. officinale* responding to far-red light, the significant enrichment of sulfur metabolism may affect the accumulation of endogenous hormones (IAA, GA, *etc.*), which in turn affects the shade-avoidance effect of *D. officinale*.

## The *D.officinale* responds to far-red light through potassium ion transmembrane transport

GO analysis of DE tagert genes revealed that potassium ion transmembrane transport was significantly enriched in the FR8-CK combination (Table 2), in which these target genes was $K^+$ transporter (*Trk/HKT*) family (*Gierth & Mäser, 2007*), including gene-MA16_Dca007090 ($K^+$ transporter 7, *HKT* 7), gene-MA16_ Dca001413 ($K^+$ transporter 11, *HKT* 11), gene-MA16_ Dca001782($K^+$ transporter 26, *HKT* 26). The target gene of novel_miR_483, gene-MA16_Dca001413 (*HKT* 11, $K^+$ transporter 11), exhibited higher expression under far-red light than under white light (Fig. 6). GO analysis of DE genes revealed that ATPase activity was significantly enriched in the FR2-CK and FR8-CK combination (Table 2). The above pathways illustrate the key role of $K^+$ in the *D. officinale* response to far-red light.

As the most abundant monovalent cation in plants, $K^+$ is one of the main osmotic regulators in the process of plant stomatal movement. Stomata are the main portals for the exchange of water and gas between plants and the environment. The opening of stomata is beneficial to the plant body to provide power for the transport of substances from the roots to the shoots through transpiration. At the same time, stomata also constitute the main ways for the photosynthesis substrate $CO_2$ to enter the plant body. The closure of stomata can prevent plants from wilting due to excessive water loss and can also prevent pathogenic microorganisms from invading plants through stomata (*Chen et al., 2017*). Plant guard cells can autonomously sense different environmental signals through multiple mechanisms and regulate ion channels/transporters in the plasma membrane and endomembrane systems through independent or intersecting signal transduction pathways, regulate the opening and closing of stomata, and balance water loss, and $CO_2$ absorption in response to the

environment (*Murata, Mori & Munemasa, 2015*). The photoreceptors on the plant guard cell membrane starts the photosynthetic light reaction after receiving the light stimulus. A large amount of ATP is produced through photosynthetic phosphorylation, which provides energy for H$^+$-ATPase continuously (*Gao, Wu & Wang, 2017*; *Hauser, Brandt & Schroeder, 2015*). At the same time, light activates the plasma membrane H$^+$- ATPase, and the H$^+$-ATPase uses the energy generated by the hydrolysis of ATP to continuously pump H$^+$ to the outside of the cell, resulting in an electrochemical potential gradient across the plasma membrane so that the mass potential-dependent influx K$^+$ channels and anion channels on the membrane open, allowing a large amount of K$^+$ to enter guard cells, and the K+ concentration in guard cells increases (*Gao, Wu & Wang, 2017*; *Hauser, Brandt & Schroeder, 2015*). Therefore, far-red light participates in stomatal movement through the transmembrane transport of potassium ions, which in turn affects the growth and development of *D. officinale*.

## MiRNAs participate in the signal transduction pathway of far-red light in the shade avoidance of *D. officinale*

On the basis of previous studies (*Zheng et al., 2013a*; *Sharkhuu et al., 2014*), by mining miRNA data, we established a regulatory network of far-red light on the shade-avoidance response of *D. officinale* (Fig. 7).

miR395b may be involved in the signal transduction pathway of far-red light on *D. officinale* through the target gene PIF3 (gene-MA16_Dca015429), and novel_miR_159 may be involved in the signal transduction pathway of far-red light on *D. officinale* through the target gene PIF4 (gene-MA16_Dca005372). PIFs belong to a subfamily of the basic helix-loop-helix (bHLH) superfamily of transcription factors, of which there are seven members: PIF1/PIF3-LIKE 5 (PIL5), PIF3, PIF4, PIF5/PIL6, PIF6/PIL2, PIF7 and PIF8. PIFs interact directly with phyB through their conserved N-terminus, termed the active phyB-binding motif (*Ren et al., 2016*). In low R:FR conditions or shade environments, PIF3, PIF4, PIF5, and PIF7 are involved in shade-avoidance responses (*Huang et al., 2019*). When plants sense low R:FR conditions, the active form of phyB is reduced, releasing the transcriptional activity of PIFs and inducing plant growth responses to shade (*Huang et al., 2019*). PIFs control the expression of these genes by directly binding to the promoter regions of cell wall-associated genes, such as xyloglucan endotransferglycosidases and xyloglucan endotransferosidases/hydrolases, which are essential for plant growth in response to shade, and no enzymes are needed (*Hornitschek et al., 2012*). There are also many growth-related genes that are indirectly regulated by PIFs (*Sun et al., 2020*). PhyB is activated and translocated into the nucleus under conditions of a high R:FR ratio, thereby inactivating PIF4, PIF5, and PIF7 and inhibiting PIF-dependent transcription. Under low R:FR ratio conditions, phyB is inactivated, which allows PIF to accumulate and regulate the transcription of its downstream targets (*Sun et al., 2020*). There are many IAA-related genes, and this process changes the expression of the AUXIN (AUX)/INDOLE-3-ACETIC ACID-INDUCIBLE (IAA) IAA signalling gene in cells through the transcription of the YUCCA (YUC) gene and then promotes IAA transport through PIN-FORMED (PIN)

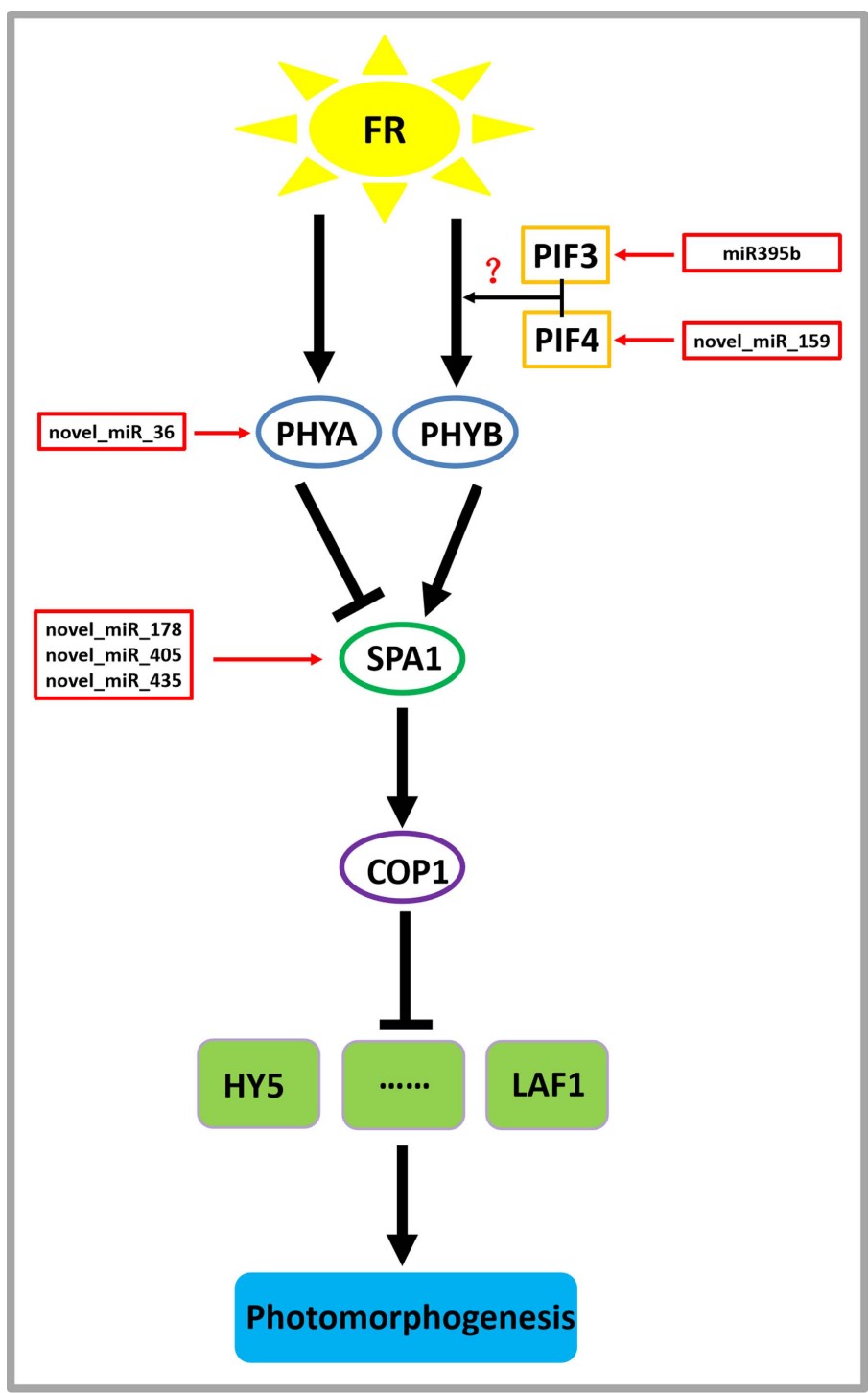

**Figure 7** **Model depicting the regulatory miRNA-mediated mechanisms of photomorphogenesis under far-red light.** FR, far-red light. The question mark indicates an unknown relationship that requires further verification.

proteins, resulting in an increase in the concentration of IAA in cells, a process that ultimately leads to cell elongation (*Sun et al., 2020*).

A novel_miR_36 may participate in the signal transduction pathway of far-red light on the shade avoidance of *D. officinale* through the target gene PHYA (gene-MA16_Dca006821) (Fig. 6). Phytochromes play an important role in the shade-avoidance response and can be divided into two categories according to their stability: photostable PHYA and photostable PHYB-PHYE (*Kong & Zheng, 2020*). PHYB is the main phytochrome involved in the regulation of the shade-avoidance response. PHYD and PHYE are positive cofactors of PHYB in the regulation of the shade-avoidance response. Under the dominant role of PHYBs, they redundantly regulate the shade-avoidance response of plants. The role of PHYC in the shade-avoidance response has not been determined. PHYA plays a decisive role in the inhibition of seed germination, de-etiolation, and hypocotyl elongation under far-red light conditions (*Lim, Park & Sukjoon, 2018*; *Shen et al., 2009*). Related studies have shown that the phyA mutant does not exhibit an obvious shade-avoidance response phenotype like that of the phyB mutant under white-light conditions, but its hypocotyls are longer than those of the wild type under simulated shade conditions. In addition, the mutations of phyA enhance the shade-avoidance response syndrome of the phyB single mutant and phyB phyD phyE triple mutant under white-light conditions, and phenotypes such as hypocotyl and petiole elongation, reduced leaf area, and decreased chlorophyll content are more obvious (*Lim, Park & Sukjoon, 2018*; *Shen et al., 2009*).

Through the target gene SPA1, novel_miR_178, novel_miR_405, and novel_miR_435 may participate in the signal transduction pathway governing the effects of far-red light on the shade avoidance of *D. officinale* (gene-MA16_Dca020963) (Fig. 7). SPA1 is a negative regulator of phyA-dependent far-red light signalling pathway-specific photomorphogenesis (*Zheng et al., 2013a*). SPA1 is also involved in the red light signalling pathway and is resistant to the inhibitory effect of phyB on hypocotyl elongation (*Zheng et al., 2013a*). SPA1 is structurally similar to another negative regulator, COP1, and is required for its full function (*Martínez, Nieto & Prat, 2018*). SPA1 and COP1 interact to form a ligase complex, which together mediate the positive regulators of the light signal transduction pathway, such as HY5, LAF1, and HFR1, which are subsequently degraded through the 26S proteosome pathway and affect the plant shade-avoidance response (*Martínez, Nieto & Prat, 2018*).

## CONCLUSIONS

This study provides the first demonstration of far-red light on microRNAs involved in the shade-avoidance response of *D. officinale* through an RNA-seq analysis. Previous studies have found that in *D. officinale* 730 nm (far-red) light can promote the accumulation of plant metabolites, increase leaf area, and accelerate stem elongation. Based on the transcriptomic, physiological and biochemical analyses, we revealed that folic acid metabolic pathway, cutin, suberin and wax biosynthesis, sulfur metabolism, and potassium ion transmembrane transport play an important role in the response of *D. officinale* to blue

lasers. Some miRNAs participate in the signal transduction pathway of far-red light in the shade avoidance of *D. officinale*. These findings will be helpful for generating new insights for the high-yield production of functional metabolites of *D. officinale*.

## ACKNOWLEDGEMENTS

We thank American Journal Experts for editing the English text of a draft of this manuscript.

### Funding
This work was funded by the Natural Science Foundation of Fujian Province (2020J01377), the 2021 National Fund Cultivation Project of Sanming University (PYT2101), the Sanming University Scientific Research Foundation for High-level Talent (18YG01, 18YG02, 19YG06), and the 2021 Special Commissioner of Science and Technology of Fujian Province. The funders had no role in study design, data collection and analysis, decision to publish, or preparation of the manuscript.

### Grant Disclosures
The following grant information was disclosed by the authors:
Natural Science Foundation of Fujian Province: 2020J01377.
2021 National Fund Cultivation Project of Sanming University: PYT2101.
Sanming University Scientific Research Foundation for High-level Talent: 18YG01, 18YG02, 19YG06.
2021 Special Commissioner of Science and Technology of Fujian Province.

### Competing Interests
The authors declare there are no competing interests.

### Author Contributions
- Yifan Yang performed the experiments, analyzed the data, prepared figures and/or tables, authored or reviewed drafts of the article, and approved the final draft.
- Yuqiang Qiu performed the experiments, prepared figures and/or tables, and approved the final draft.
- Wei Ye conceived and designed the experiments, performed the experiments, authored or reviewed drafts of the article, and approved the final draft.
- Gang Sun conceived and designed the experiments, authored or reviewed drafts of the article, and approved the final draft.
- Hansheng Li conceived and designed the experiments, analyzed the data, authored or reviewed drafts of the article, and approved the final draft.

### Data Availability
The sequencing data of D. officinale under the different light treatments is available in the National Genomics Data Center (NGDC) Sequence Read Archive: PRJCA010065.

The reference genome version of Dendrobium officinale in this article was updated on April 11, 2019: https://www.ncbi.nlm.nih.gov/genome/?term=txid906689[orgn].

## Supplemental Information

Supplemental information for this article can be found online at http://dx.doi.org/10.7717/peerj.15001#supplemental-information.

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
