# Peer review of "RNA sequencing-based exploration of the effects of far-red light on microRNAs involved in the shade-avoidance response of D. officinale"

_PeerJ, doi:10.7717/peerj.15001_

## Round 0.1 · original submission · Minor Revisions

Dear Dr Li,

Thank you for your submission to PeerJ. It is my opinion, having studied the reviews, that it requires a number of Minor Revisions.

Reviewer 1 ·

Basic reporting

The Manuscript entitled, “RNA sequencing-based exploration of the effects of far-red light on microRNAs involved in the shade avoidance response of D. officinale” highlighted the role of micro RNA regulatory effects on secondary metabolite production in the far-red light effect under shade avoidance response in Dendrobium offcinale.
Author Yang et al. depicted the significant work to speculate the more effect of far red light response compared to red light and blue light on the several novel micro RNA responses over different target gene expression, mostly related to metabolites Apart from this they have highlighted the possible pathways e.g. Wax, cutin and suberine, folate biosynthesis and sulfur metabolism, etc. due to the far red light response in shade avoidance condition.

Experimental design

Although all the results are analyzed on computational software program (Bioinformatics tools) except the qPCR data on target gene expression of micro RNA, ascorbic acid, and folic acid content.
I conclude that all the experimental data are significant to deciphering the conclusive story of metabolite production under far-red light response in shade avoidance conditions.
Still few of the points have to be pointed out by the authors to justify their work

Validity of the findings

No Comment

Additional comments

#Comment 1 Under the material and method section (Line number 107) the intensity of light is mentioned as 200 µmol/m2/s which is a little higher intensity of light compared to normal culture room conditions. More than 150 µmol/m2/s may create photo-oxidative stress as the plant is kept for 60 days longer period.
How can you justify this light intensity?
This concern is important as it might be possible that some of the micro RNA are responding because of high intensity irrespective of the light wavelength.

#Comment 2 In line number 276-278 it is mentioned that far-red light is responsive for K+ transmembrane transport, so there are three transporter families of K+ are well known- K(+) transporter (Trk/HKT) family, K(+) uptake permeases (KT/HAK/KUP) and cation proton antiporters (CPA).
So now According to your findings which one family member is more effectively responding under far-red light response? Can you please explain?
Reference- (Gierth, M., & Mäser, P. (2007). Potassium transporters in plants–involvement in K+ acquisition, redistribution and homeostasis. FEBS letters, 581(12), 2348-2356.)

#Comment 3 In the section “QPCR analysis of DE miRNAs and their target gene expression” the author has stated that “Novel_miR_53 targets gene-MA16_Dca007605, 321 miR395b targets gene-MA16_Dca003285, novel_miR_36 targets gene-MA16_Dca020471, and 322 novel_miR_159 targets gene-MA16_Dca005372. Only these 4 pairs of miRNAs and target gene 323 expression patterns were negatively correlated”. It is explained that some other micro RNA might also play a role in this negative correlation. Is it also possible that this micro RNA is upregulating the expression of inhibitors? Which are downregulating the expression of these target genes.

#Comment 4 Author has explained that in Line number 434-437- “ATPase activity was significantly enriched in the FR1-CK and FR4-CK combination”. Does it related to H+ATPase pumps as it is again highlighted in line number 452-457 which help to drive the K+ transporter inside the guard cell?
I raise the concern of light effect on the phosphorylation process, as H+ATPase gets phosphorylated. As Phosphorylation is an essential key process for ATPase activity. Overall does far red light stimulate the phosphorylation process rather than direct effect on the H+ATPase pump?
Can you justify this point?
Reference- Falhof, J., Pedersen, J. T., Fuglsang, A. T., & Palmgren, M. (2016). Plasma membrane H+-ATPase regulation in the center of plant physiology. Molecular Plant, 9(3), 323-337.

Suggestions:
Network analysis Figure 4 is very hazy or blurred, the resolution should be improved, and none of the gene names cannot be read out in this figure. The Author should redraw this image.

In Line Number 400- The word needs to be corrected, it should be both not Toth.

Reviewer 2 ·

Basic reporting

The manuscript titled as “RNA sequencing-based exploration of the effects of far-red light on microRNAs involved in the shade-avoidance response of D. offcinale” investigates the role of microRNAs in regulation of shade-avoidance mechanism of D. offcinale. Though the manuscript is interesting, and findings may help the scientific community to further explore the new suggested target genes, here are some following suggestions and queries:
1. In material and methods section, please write what is the standard working solution (line 162). Mention the name of kit and manufacturer in line 164. If the standard working solution is from the kit then rewrite this section mentioning the kit and manufacturer’s name in starting.
2. Similarly, for ascorbic acid determination, write the kit and manufacturer’s name in the starting.
3. Please mention if these tests are spectrophotometric or colorimetric.
4. Line 174-175 should be converted to past tense.
5. Please mention the name and accession number of reference gene used for qRT-PCR experiments.
6. Please readjust the data in Table 2, as it is confusing to identify that how many GO terms are present in a particular category (Biological process, cellular component and molecular function).
7. Instead of providing the scattered information along the whole discussion section, I suggest that figure 7 showing the Model depicting the regulatory miRNA-mediated mechanisms of photomorphogenesis under far-red light, should be discussed in detailed paragraph with current new findings supported by the citation of relevant available literature.

Experimental design

No comment

Validity of the findings

No comment

Additional comments

No comment

Reviewer 3 ·

Basic reporting

No comment

Experimental design

No comment

Validity of the findings

No comment

Additional comments

In the manuscript entitled "RNA sequencing-based exploration of the effects of far-red light on microRNAs involved in the shade-avoidance response of D. officinale", the authors have demonstrated that miRNAs participate in the signal transduction pathway of far-red light in the shade avoidance of D. officinale. The comments and suggested changes are provided below that might improve the manuscript.

Comments and suggestions:
1. In line-80, ‘Zhou et al.’ can be cited in the text like ‘Zhou et al. (2016)’. No need to repeat the citations. Similarly in line-84, Yan et al. can be modified.
2. In line-108, ‘55%~60%’ can be modified to ‘55%-60%’.
3. There are inconsistencies in the presentation of light groups; red:blue:far-red (40:40:120) is denoted as FR4 in Materials & Methods section (line-115) whereas in Results section it is written FR2 (line 195, 201) and in Table 1, Table S3, Table S4 it is mentioned as FR8.
4. Instead of transcriptome sequencing in line-123, small RNA sequencing will be more appropriate.
5. In line-132, it is written ‘Bowtie v2.2.3 software was used’. For what? The sentence is also incomplete. Please correct it.
6. Please correct the sentence ‘The remaining reads were used to detect known miRNAs and novel miRNAs predicted by comparison with the genome and known miRNAs from miRBase 16.0’ (line 135-137).
7. In line-157, ‘Nr’ is repeated twice. Please correct it.
8. For Folic acid determination what was the standard working solution (line-162)? Please explain.
9. Is the kit mentioned in line-164 and line-172 same? If yes, then the description about the kit should be at the first place.
10. What is TMB in line-171. Authors need to mention the full name.
11. The methodology for Ascorbic acid determination is not written properly (line 174-178). Please write correctly.
12. In line-229, all 3 combinations are written same (needs to be corrected) whereas Fig 3D shows right combinations.
13. qPCR results are shown in Fig. 6 and Table S9. Hence it needs to be corrected in line-319-320.
14. Fig. 1 as mentioned in line-348, 373 is not showing wax content. Where is this data? What physiological experiments have been performed in this study?
15. In line-38, 365, 387, 429, 466 what are those numbers at the end of the sentence?
16. In line-382, Huang et al. showed that… is written, but other reference has been put at the end of the sentence.
17. ‘Fig. 4’ needs to be changed to ‘Fig. 3’ in line-395, 397, 398 since KEGG pathways shown in Fig. 3.
18. ‘Toth’ should be changed to ‘Both’ in line-400 and Fig. 7 to Fig. 6 in line-403.

---

## Round 0.2 · Minor Revisions

Dear Dr Li,

There are minor corrections that need to be answered before the manuscript is sent for publication.

Comments from Editor:
''Creating sequence names and illustrating them in a figure has no value unless the actual sequences are produced and made available. Charts and Venn diagrams are nice summaries, but supplemental data containing the sequence data is needed to accompany the explanation. The sequences grouped into the DE categories need to be explicitly made available, the pointer to the raw data repository would require the reader to reproduce the data analysis for which their result may differ; thus it is important that the evaluated and annotated data be produced. A table including the sequence, the category, and the annotation can be made available as supplementary data''.

Reviewer 1 ·

Basic reporting

The Revised Manuscript entitled, "RNA sequencing-based exploration of the effects of far-red light on microRNAs involved in the shade-avoidance response of D. officinale" is well addressed for the queries/comments by the author Yang et. al.

Now, this manuscript has fulfilled the relevant information based on their objectives and experimental pieces of evidence with explanations.

I highly appreciate the author's effort in concerning the comments/queries and addressing all the queries in a proper way with suitable changes in the revised manuscript.

I strongly recommend this manuscript be published in PEER J.

Experimental design

NO SPECIFIC COMMENTS

Validity of the findings

NO SPECIFIC COMMENTS

Additional comments

RECOMMENDED FOR PUBLICATION

Reviewer 3 ·

Basic reporting

no comment

Experimental design

no comment

Validity of the findings

no comment

Additional comments

Authors have answered most of my comments and suggestions. I am happy to recommend this paper for publication.

---

## Round 0.3 · Minor Revisions

Dear Authors,

The Section Editor, has commented and said:

"The additional tables were helpful in documenting the observed categories; however, the is no sequence data seen or available to back up the novel sequence names provided. I presume the data at the NGDC SRA repository only has raw data and is not connected to the sequence names listed in the provided supplements. The SRA project PRJCA010065 was not accessible to determine this. The manuscript in its current state has no connection to any navigatable sequence data. Though additional clarifications were made, the manuscript still needs revision to rectify this point of pointing to tractable sequence data. This was not apparent in the most recent revision."

Authors are requested to relook into the above comments and submit the revised manuscript for further processing.

wish you good luck
Ratnakumar Pasala

---

## Round 0.4 · Minor Revisions

Dear Dr. Li,

Thank you for your submission to PeerJ.

Gerard Lazo, the Section Editor, has commented and said:
"I have visited the NGDC site and have tried to access the sequence data there. The sequence data available is still in a raw form only being exposed as a fastq (fq,gz) type file which would require processing; where is the fasta file with clear annotation of the members?. Saying the data is available at a site and clearly pointing to what you are presenting are different matters. If the authors are going to make it so difficult for the author to access the data they are discussing, there needs to be an explanation within the manuscript explaining how to access the data for the readership. The clearer the better. I did not see anything like the author describes in the rebuttal; an exact link may be needed."

Therefore, I request you go through the comments carefully and respond accordingly.

With best regards
Ratan Kumar

---

## Round 0.5 · accepted · Accept

Dear Dr. Li,

Thank you for your submission to PeerJ.

I am writing to inform you that your manuscript - RNA sequencing-based exploration of the effects of far-red light on microRNAs involved in the shade-avoidance response of D. officinale - is accepted for publication now that you have provided the annotated FASTA files.